# Dual Semi-Supervised Learning for Classification of Alzheimer’s Disease and Mild Cognitive Impairment Based on Neuropsychological Data

**DOI:** 10.3390/brainsci13020306

**Published:** 2023-02-10

**Authors:** Yan Wang, Xuming Gu, Wenju Hou, Meng Zhao, Li Sun, Chunjie Guo

**Affiliations:** 1Key Laboratory of Symbol Computation and Knowledge Engineering of Ministry of Education, College of Computer Science and Technology, Jilin University, Changchun 130012, China; 2Department of Neurology and Neuroscience Center, The First Hospital of Jilin University, Changchun 130021, China; 3Department of Radiology, The First Hospital of Jilin University, Changchun 130021, China

**Keywords:** Alzheimer’s disease, semi-supervised Learning, neuropsychological test, difference regularization

## Abstract

Deep learning has shown impressive diagnostic abilities in Alzheimer’s disease (AD) research in recent years. However, although neuropsychological tests play a crucial role in screening AD and mild cognitive impairment (MCI), there is still a lack of deep learning algorithms only using such basic diagnostic methods. This paper proposes a novel semi-supervised method using neuropsychological test scores and scarce labeled data, which introduces difference regularization and consistency regularization with pseudo-labeling. A total of 188 AD, 402 MCI, and 229 normal controls (NC) were enrolled in the study from the Alzheimer’s Disease Neuroimaging Initiative (ADNI) database. We first chose the 15 features most associated with the diagnostic outcome by feature selection among the seven neuropsychological tests. Next, we proposed a dual semi-supervised learning (DSSL) framework that uses two encoders to learn two different feature vectors. The diagnosed 60 and 120 subjects were randomly selected as training labels for the model. The experimental results show that DSSL achieves the best accuracy and stability in classifying AD, MCI, and NC (85.47% accuracy for 60 labels and 88.40% accuracy for 120 labels) compared to other semi-supervised methods. DSSL is an excellent semi-supervised method to provide clinical insight for physicians to diagnose AD and MCI.

## 1. Introduction

Alzheimer’s disease (AD) is a neurodegenerative brain disease, which indicates that the condition gradually worsens over time. Patients in the early stages of AD, namely mild cognitive impairment (MCI), have a greater likelihood of converting to AD years later [1]. The lesions of the disease occur mainly in the cerebral cortex and hippocampus, which causes patients to develop cognitive impairments in language, memory, and other aspects [2]. Positron emission tomography (PET), magnetic resonance imaging (MRI), and cerebrospinal fluid (CSF) biomarkers are included in A/T/N system for research [3], which highlights the importance of reliable biomarkers for AD diagnosis. However, these measures’ high cost and intrusiveness limit their widespread application and potential in clinical screening patients for AD [4]. Therefore, it is vital to identify non-invasive, reliable, and widely available diagnostic biomarkers for AD.

Research has suggested that the traditional diagnosis of cognitive disorders remains limited to subjective symptoms and observable features, and that ML offers a novel paradigm that can enable automated and more objective evaluation of various psychiatric diseases [5]. In recent years, researchers have used machine learning (ML), especially deep learning (DL), instead of traditional methods to assist in the diagnosis of AD [6,7,8]. In particular, the fully supervised DL-based method is the dominant approach in AD diagnosis. Specifically, convolutional neural networks (CNN) and graph convolutional networks (GCN) have demonstrated excellent performance in medical image classification tasks [9]. Amini et al. [10] compared several ML methods for AD diagnosis using functional magnetic resonance imaging (fMRI) images. They showed that CNN outperformed all other traditional ML techniques in effectively detecting AD severity. Zhou et al. [11] proposed an interpretable GCN framework using multimodal brain imaging data to classify AD, MCI, and normal controls (NC). Considering the node features and their connectivity in the network, Zhou et al. [12] further proposed a sparse interpretable GCN framework, which uses multiple modalities of brain imaging data to classify AD. However, due to the complexity of disease pathology, it is costly to obtain the ground truth labels for AD and MCI, which requires expert knowledge. The lack of labeled data remains a significant obstacle to the progress of DL in AD diagnosis [13]. Semi-supervised learning (SSL) methods in DL are particularly suitable for situations where labeled data is scarce [14].

Neuropsychological tests are commonly used in clinical practice to determine the degree of cognitive impairment including AD and MCI [15]. These tests are short-cycle, low-cost, and easy to conduct compared to medical imaging and CSF measures. Research suggests neuropsychological test results may have as much screening potential for AD patients as CSF and MRI biomarkers [16]. Grassi et al. [17] used predictors integrating sociodemographic characteristics, cognitive measures, clinical tests, etc. They used multiple supervised learning methods to identify which subjects with MCI would convert to AD in the following years. Battista et al. [18] used a combination of support vector machine (SVM) and 131 measures from 324 participants, including different neuropsychological tests to classify subjects with different clinical dementia ratings (CDR). Although ML methods such as SVM have yielded promising results, no predictors can be used as the gold standard, and some studies have found problems with some measures [19]. As advanced and prevalent ML methods, neural networks have rarely been applied to diagnosing AD using neuropsychological tests, whose widespread application will provide clinical insight for physicians to determine the degree of cognitive impairment.

To address the problem of difficulty in obtaining labeled data, this paper proposes a new method for Alzheimer’s disease classification that reduces the need for labeled data based on SSL. Our proposed method applies easily available and non-invasive neuropsychological test data for the diagnosis of AD. First, we calculate the correlation of each neuropsychological test on the diagnostic results by Pearson’s correlation coefficient and select features according to the magnitude of the coefficients. Then, we propose the dual semi-supervised learning (DSSL) algorithm, which uses two different encoders to learn different feature representations of the samples. In addition, we combine pseudo-labeling with consistency regularization. The two predictions obtained from the two feature representations are hard-labeled and then used as mutual pseudo-labels.

To evaluate the classification performance of DSSL, we conduct extensive experiments in the Alzheimer’s Disease Neuroimaging Initiative (ADNI) database (http://ADNI.loni.usc.edu/ (accessed on 18 December 2021)). Experimental results show that DSSL largely outperforms existing semi-supervised methods in a variety of evaluation metrics, and the short training time of the model demonstrates its practicality in clinical diagnosis. By dividing the training data and performing training many times, we find that DSSL also has strong stability.

The contributions of this paper are:We select some neuropsychological tests by feature selection, which are better predictors of automatic classification and can provide clinical diagnostic references to physicians;Propose a novel semi-supervised method that introduces difference regularization in unsupervised loss computation to enhance model perturbations by learning two different feature representations;Propose a tri-classification framework for cognitive impairment based on improved SSL and CNN, which identifies AD, MCI, and NC using the most straightforward method (i.e., neuropsychological tests) and fewer labels. Experimental results based on the ADNI dataset indicate that the classifier outperforms other semi-supervised methods in terms of accuracy and stability.

## 2. Theoretical Backgrounds

Deep neural networks contain many hidden layers, each containing a large number of hidden nodes, which gives it a powerful fitting capability to approximate almost any complex function. However, the powerful fitting ability of deep learning relies on a large amount of training data. Training with only a small amount of labeled data often leads to overfitting problems [13]. In addition, the interpretability of deep learning is still being explored by researchers [20].

### 2.1. Semi-Supervised Learning

SSL is a powerful method for training models on large datasets with only a small number of labels. SSL alleviates the need for labeled data by learning the connections and differences between unlabeled data. In the following sections, we discuss the background related to this work. For a three-class classification problem, let (x,p) denote a labeled example and *u* denote an unlabeled example, respectively. *D* denotes the number of labeled samples and $\mu_{D}$ denotes the number of unlabeled samples. Let $p_{model}\left(y|x\right)$ denote the predicted probability generated by the model with input *x*. Let I(condition) denote 1 if the condition holds and 0 if not. Let H(p,q) denote the cross-entropy between two probability distributions *p* and *q*.

In the semi-supervised task, we aim to predict the classification using several image labels accurately. Especially for a reasonably large dataset, labeling these images manually could be a tedious and challenging task. Therefore, it is now understandable why we chose the semi-supervised algorithm for our study.

#### 2.1.1. Consistency Regularization

Consistency regularization is an essential component of the deep neural network model in the SSL algorithm. Consistency regularization employs a perturbation strategy in which the same sample is altered to yield various outputs. The perturbation approach assumes that the model should output similar predictions when the same input sample is perturbed. This idea was first proposed in [21] and promoted by [22,23]. The perturbation methods can be divided into sample perturbation methods and model perturbation methods according to the different perturbation stages. Sample perturbation refers to the data augmentation of the input sample to obtain a new sample that is different from the previous sample but mostly similar; model perturbation is a change in the model, where the same sample undergoes a different model to produce a difference in the output results. Consistency regularization in the model is mainly trained on unlabeled data by the loss function:(1)$${∥p_{model}\left(y\left|A\left(u\right)\right.\right)-p_{model}\left(y\left|A\left(u\right)\right.\right)∥}_{2}^{2},$$
where ${∥\cdot ∥}_{2}$ denotes the L2 norm and A(u) denotes data augmentation. Note that both A(u) and $p_{model}$ are random functions, so the two terms in Equation (Equation 1) are not the same. The consistency regularization with different A(u) belongs to the sample perturbation method. Virtual adversarial training [24] (VAT) uses adversarial perturbation to generate an adversarial sample that forms a difference from the original sample, and MixMatch [25] uses the mixup [26] method to perform data augmentation on the input samples. FixMatch [27] uses both strong and weak augmentations, and experiments with strong augmentations based on RandAugment [28] and CTAugment [29]. Most of the existing sample perturbation methods, however, are data augmentation methods used for image data. It is not widely applicable to other types of data. The consistency regularization with different $p_{model}$ belongs to the sample perturbation method. Π-model [23] uses the randomness of dropout [30] to perturb the model so that the outputs of the same input sample are different. Temporal ensembling [23] uses the average of previous model checkpoints when generating artificial labels for comparison with the current prediction. Mean teacher [31] divides the model into two types: the student model, which is a general training model, and the teacher model, which is obtained by an exponential moving average of the parameters of the student model. For the same input, the different outputs obtained by the student and teacher models constitute consistency regularization.

#### 2.1.2. Pseudo-Labeling

The low-density assumption is a common fundamental assumption in SSL, referring to the classification boundary not passing through high-density regions in the input space. One way to achieve this assumption requires SSL models to output low-entropy predictions for unlabeled data. Pseudo-labeling [32] implicitly minimizes entropy by generating a hard (one-hot) label on the high-confidence prediction results of unlabeled data and using this hard label along with the model prediction result as parameters for the standard cross-entropy loss. Letting $q=p_{model}\left(y|u\right)$ and $\hat{q}=arg\;\max\left(q\right)$, the loss function used for the pseudo-labeling can be expressed as:(2)$$I\left(\max\left(q\right)\geq \tau \right)H\left(\hat{q},q\right),$$
where τ denotes the threshold. Pseudo-labeling treats the predictions of SSL classifiers on unlabeled data as artificial labels.

#### 2.1.3. Label Propagation

Label propagation is a graph-based SSL method that associates all labeled and unlabeled samples by constructing a graph. The nodes in the graph include labeled and unlabeled samples, and the weights of the edges represent the similarity between two nodes. The labels of the samples are propagated through the edges between the nodes. Recently, it has been combined with pseudo-labels as a novel way of giving pseudo-labels or calculating losses based on pseudo-labels. Iscen et al. [33] used a label propagation method based on the manifold assumption to predict the current node based on the k nodes with high similarity, and used the predicted results to generate pseudo-labels for unlabeled samples. SimPLE [34] introduces pair loss in addition to supervised loss and consistency loss, which decrease the noise of pseudo-labels by setting a confidence threshold and similarity threshold.

### 2.2. Contrastive Learning

Self-supervised learning, unlike supervised learning which requires expensive labeling, is able to use unlabeled data to learn the underlying representation. Contrast learning, one of the important methods of self-supervised learning, aims to learn an encoder that encodes data of the same kind similarly and makes the encoding results of different classes of data as different as possible. The Pretext task is a self-supervised task using pseudo-labels to learn data representation. How to design the pretext task to better fit the SSL downstream tasks is the key to incorporating self-supervised learning into the SSL model. The CCSSL [35] framework introduces class-aware contrast loss on top of the SSL model, seamlessly integrating clustering and comparison in the feature space. LaSSL [36] learns differentiated feature representations that enable aggregation of same-class samples and dispersion of different class samples by minimizing class-aware contrast loss and performs label propagation based on the feature representations.

## 3. Materials and Methods

### 3.1. ADNI Database

Data used in this study is obtained from the ADNI database. ADNI was launched in 2003 as a longitudinal multicenter study led by Principal Investigator Michael W. Weiner. The initial objective of ADNI was to develop MRI, PET, and other biomarkers for early detection and tracking. For up-to-date information, see www.adni-info.org (accessed on 18 December 2021). In this study, we chose baseline neuropsychological data from the preliminary phase of the project (ADNI-1). The data we used are from 819 subjects including 188 AD subjects, 402 MCI subjects, and 229 NC subjects. The characteristics of the subjects selected for this study are shown in Table 1.

### 3.2. Neuropsychological Data

The itemized scores of seven neuropsychological tests are used, including the Alzheimer’s disease assessment scale-cognitive (ADAS-Cog) [37], the mini-mental state exam (MMSE) [38], the clinical dementia rating (CDR) [39], the Rey auditory verbal learning test (RAVLT) [40], the functional activity questionnaire (FAQ) [41], the neuropsychiatric inventory Q (NPIQ) [42], and the geriatric depression scale (GDS) [43]. These neuropsychological tests are widely used to determine the degree of cognitive impairment in clinical settings. Section A.1 details the cognitive functions associated with each test. A total of 64 itemized scores are derived from these seven tests. For each test, we use a different number of sub-scores, including 15 rubric scores from ADAS-cog, 31 rubric scores from MMSE, 1 rubric score from CDR, 4 rubric scores from RAVLT, 11 rubric scores from FAQ, 1 rubric score from NPIQ, and 1 rubric score from GDS. In the semi-supervised learning task of this paper, each itemized score is considered a feature of the sample. We provide a brief introduction of the neuropsychological tests selected as features in Section A.2.

### 3.3. Method

#### 3.3.1. Features Selection

Feature selection has a highly important role in DL. Pearson’s correlation coefficient (PCC) [44], one of the most common feature selection methods, is applied to neuropsychological tests in this study. Although PCC cannot assess how similar a combination of multiple variables is to a single variable, it is still the most popular method for calculating the similarity between two variables. PCC evaluates the degree of correlation between two variables by calculating the standard deviation of the two variables and the covariance between them. PCC between the two variables *X* and *Y* is defined as:(3)$$\rho_{X,Y}=\frac{COV\left(X,Y\right)}{\sigma_{X}\sigma_{Y}}=\frac{E\left[\left(X-\mu_{X}\right)\left(Y-\mu_{Y}\right)\right]}{\sigma_{X}\sigma_{Y}},$$
where COV denotes the covariance, $\mu_{X}$ denotes the mean of *X*, $\mu_{Y}$ denotes the mean of *Y*, $\sigma_{X}$ denotes the standard deviation of *X*, $\sigma_{Y}$ denotes the standard deviation of *Y*, and *E* denotes the expectation. The value calculated by Equation (Equation 3) varies from −1 to 1. A value between 0 and 1 denotes that the two variables are positively correlated, while a value between −1 and 0 denotes that they are negatively correlated. The closer the absolute value is to 1, the stronger the correlation between the two variables.

#### 3.3.2. Dual Semi-Supervised Learning

In this subsection, we introduce DSSL, a novel semi-supervised method, as a convenient and accurate classifier for the clinical diagnosis of AD. Inspired by fixMatch [27], DSSL combines consistency regularization and pseudo-labeling, two SSL methods discussed in the previous section. Figure 1 shows the overall view of the model for the supervised and unsupervised parts. DSSL applies model perturbation through two different encoders. To make the two encoders learn as different features as possible, DSSL introduces difference regularization, which stretches the distance between the features extracted from the input by the two encoders. The network architecture of the encoders is shown in Figure 2. For a sample, two different feature vectors are obtained through Encoder1 and Encoder2, respectively. These two vectors are then fed into the multilayer perceptron (MLP) network to obtain two prediction results. They serve each other as pseudo-labels for the different prediction results, which constitutes consistency regularization. Algorithm 1 provides the complete DSSL algorithm.
**Algorithm 1** DSSL algorithm.
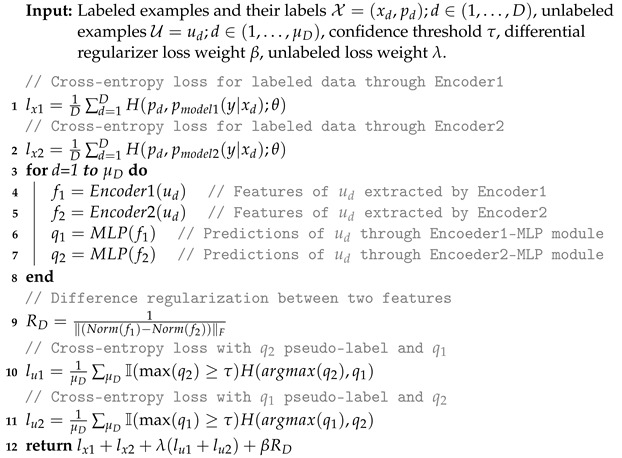


#### 3.3.3. Regularization of DSSL

Two regularizations are introduced in our approach, a difference regularization so that the two encoders learn different features, and a consistency regularization combined with pseudo-labeling.

**Differential Regularizer ($R_{D}$)**—We expect to learn two different aspects of the feature representation from Encoder1 and Encoder2. Therefore, we apply a difference regularization between the two features output by the two encoders. The distance between the two feature vectors is appropriately widened to increase the perturbation and to prepare for the consistency regularization later. The concrete implementation is shown below:(4)$$R_{D}=\frac{1}{{∥\left(Norm\left(f_{1}\right)-Norm\left(f_{2}\right)\right)∥}_{F}},$$
where Norm is the normalization operation, which aims to put two feature representations into an order of magnitude to compare, $f_{1}$ and $f_{2}$ are the feature vectors learned by the two encoders, and ${∥\cdot ∥}_{F}$ denotes the Frobenius norm.

**Consistency Regularization**—DSSL combines consistency regularization with the pseudo-labeling approach by turning the model’s predictions into hard labels. Not all hard labels of the samples are involved in the operation as parameters of the model’s loss function. The model keeps only the pseudo-labels whose maximum prediction probability is higher than a predefined threshold. Assuming $q_{2}=p_{model2}\left(y|u\right)$, where $p_{model2}$ is the prediction of the Encoder2-MLP module, and $q_{2}$ is the prediction probability. Similarly, $p_{model1}$ is the prediction of the Encoder1-MLP module, and $q_{1}$ is the prediction probability. We use ${\hat{q}}_{2}=arg\;\max\left(q_{2}\right)$ as a pseudo-label. In other words, the category with the highest prediction probability is obtained as the pseudo-label of the sample. More specifically, consistency regularization is defined as:(5)$$l_{u1}=\frac{1}{\mu_{D}}\sum_{\mu_{D}}I\left(\max\left(q_{2}\right)\geq \tau \right)H\left({\hat{q}}_{2},p_{model1}\left(y|u\right)\right),$$
where $l_{u1}$ is the consistency loss of $q_{1}$ with ${\hat{q}}_{2}$ as the pseudo-label, τ is a scalar hyper-parameter representing the threshold value used to determine which samples participate in calculating the loss function.

#### 3.3.4. Loss Function of DSSL

The training objective of DSSL is to minimize the following total objective function:(6)$$l_{T}=l_{x1}+l_{x2}+\lambda \left(l_{u1}+l_{u2}\right)+\beta R_{D},$$
where λ and β are regularization coefficients. $l_{u2}$ is similar to $l_{u1}$, which computes the cross-entropy loss of the hard label of $q_{1}$ with $q_{2}$. $l_{x1}$ and $l_{x2}$ are the standard cross-entropy loss between the true labels and the output of the Encoder1-MLP module, the Encoder2-MLP module, respectively. $l_{x1}$ is formulated as the following expression:(7)$$l_{x1}=-\frac{1}{D}\sum_{D}y\log\;p_{model1}\left(y|x\right),$$
where *x* is the labeled data and *y* is the accurate label of the data. Since $l_{x2}$ is similar to $l_{x1}$, it will not be discussed further here.

## 4. Results

### 4.1. Features Selection

We select a total of 64 itemized scores from 7 neuropsychological tests. To find characteristics that significantly discriminate Alzheimer’s disease, we do PCC calculations between their scores and labels. Then, the correlation coefficients are ranked in descending order of absolute value, and the top 15 features are selected as input for the subsequent semi-supervised experiments. Their corresponding PCCs are shown in Table 2. The table shows that their total scores correlate more strongly with the degree of cognitive impairment compared to the sub-scores of each test.

### 4.2. Implementation

To determine the optimal parameters of the DSSL framework, we use 5-fold cross-validation, i.e., the dataset is randomly divided into 5 folds. Each time, one fold is selected for testing and the remaining 4 folds are used for training. DSSL uses the adam optimizer to optimize the model parameters. As with FixMatch [27], we use an exponential moving average of the parameters with a decay of 0.999 to update the model instead of the decay learning rate. This allows the model to converge more smoothly at a higher number of iterations and improves the accuracy of the final prediction results [31]. Since we consider supervised loss and consistency loss to be equally important, we set the consistency regularization coefficient λ to 1.

In our implementation, the confidence threshold τ in the DSSL loss function plays a key role in the classification accuracy. To determine the optimal value of τ, we conduct experiments in which τ is varied from 0 to 0.99. To better understand the role of confidence threshold in DSSL, we refer to two measures proposed in the FixMatch approach: impurity rate (the prediction error rate of samples exceeding the threshold) and passing rate (the number of instances above the threshold as a percentage of the total), calculated as follows:(8)$$\mathrm{impurity}\;\mathrm{rate}=\frac{\sum_{d=1}^{D}I\left(\max\left(q_{1}\right)\geq \tau \right)I\left(y_{d}\neq {\hat{q}}_{1}\right)}{\sum_{d=1}^{D}I\left(\max\left(q_{1}\right)\geq \tau \right)},$$
(9)$$\mathrm{mask}\;\mathrm{rate}=\frac{1}{\mu_{D}}I\left(\max\left(q_{1}\right)\geq \tau \right).$$

Table 3 shows the quantity and quality of pseudo-labels and the DSSL classification accuracy at different τ in the 60-label case. From the results, we can see that there is a positive correlation between these two indicators, i.e., when the sample pass rate increases, the impurity rate also increases, which is in line with our expectation. Next, to determine the optimal value of the difference regularization coefficient β, we report the accuracy scores for multiple selected values of this parameter at 60 labels in Figure 3a. It can be seen that the proposed method achieves high prediction accuracy (over 82%) for different values of β, where the highest accuracy is obtained for β = 2. We also experiment with the performance variation of DSSL when trained using different training set sizes. In this experiment, we keep the number of samples with labels below 40% of the number of samples in the training set. As can be seen in Figure 3b, the performance of DSSL gradually improves as the training data increases and plateaus after the size of the training data exceeds 500.

### 4.3. Results of Disease Classification

To evaluate the performance of the SSL method, five evaluation metrics are chosen: Accuracy, Sensitivity, Specificity, Recall, and F1-score. The true positive (TP), true negative (TN), false positive (FP), and false negative (FN) rates are each related to these factors. The definitions of these evaluation measures are provided below:(10)$$\mathrm{Accuracy}\;\left(\mathrm{ACC}\right)=\frac{\mathrm{TP}+\mathrm{TN}}{\mathrm{TP}+\mathrm{TN}+\mathrm{FP}+\mathrm{FN}},$$
(11)$$\mathrm{Sensitivity}\;\left(\mathrm{SEN}\right)=\frac{\mathrm{TP}}{\mathrm{TP}+\mathrm{FP}},$$
(12)$$\mathrm{Specificity}\;\left(\mathrm{SPE}\right)=\frac{\mathrm{TN}}{\mathrm{TN}+\mathrm{FP}},$$
(13)$$\;\mathrm{Recall}\;\left(\mathrm{REC}\right)=\frac{\mathrm{TP}}{\mathrm{TP}+\mathrm{FN}},$$
(14)$$F1-\mathrm{score}\;\left(F1\right)=\frac{2\mathrm{TP}}{2\mathrm{TP}+\mathrm{FP}+\mathrm{FN}}.$$

To compare the effect of different labeled sample sizes in the training set on the classification performance, our experiments are designed with two labeled sample sizes: 60 labeled and 120 labeled. It should be noted that the rest of the training data are unlabeled samples. In the proposed model, the test data achieved an accuracy of 85.47% with 60-label training and 88.40% with 120-label training. Figure 4 shows the prediction results for the test set samples, where the boxes indicate the actual labels of the samples and the dots indicate the prediction results of the samples by DSSL. T-distributed stochastic neighbor embedding (t-SNE) can reduce high-dimensional data to two or three dimensions for data visualization. As shown in the figure, most of the sample points predicted by DSSL fall correctly in the boxes of the authentic samples.

The architecture of the two encoders in the DSSL model significantly impacts the results of the semi-supervised experiments. Figure 2 depicts the internal structure of the encoders. We experimentally test the effect of changing the encoder structure on the classification performance, especially when Encoder1 and Encoder2 have the same structure. The changes to the encoder are mainly focused on the pooling layer, applying max pooling and average pooling. Table 4 compares the classification results of the DSSL framework applying different combinations of encoders. It can be seen that the DSSL with different structures of Encoder1 and Encoder2 has better classification results.

### 4.4. Comparison with Other Methods

We compare our proposed method with other existing semi-supervised methods. The five methods described in Section 2: MixMatch [25], FixMatch [27], SimPLE [34], CCSSL [35], and LaSSL [36] are considered as baseline methods. To fairly compare these methods, we reimplement them using the same deep learning framework (i.e., PyTorch) and model. Considering that the strong augmentation part of the baseline methods is only applicable to image data, we choose mixup [26] as an alternative to RandAugment [28] or CTAugment [29] for data augmentation. Table 5 compares the performance of all baselines and DSSL. We compute the evaluation results for both cases with labeled samples of 60 and 120. All results are averaged for the 5-fold cross-validation. It can be seen that DSSL outperforms all baselines to a large extent, both in the 60-label and 120-label cases. Figure 5 illustrates box plots of the accuracy of the 5-fold cross-validation experiments for the cases of 60 and 120 labels, respectively.

Although we achieve the best classification results in the 5-fold cross-validation experiments, the selection of different labeled data can seriously affect the classification performance for the SSL algorithm. We randomly select labeled samples from the training set and repeat this process 100 times to obtain 100 division results. We train these 100 divisions sequentially to observe the stability of the algorithm. The variance of the 100 times predictions for the DSSL and each baseline are shown in Table 6. For visualization purposes, we select the three models with the slightest variance in each of the two cases and plot their 100 times results as line graphs, as shown in Figure 6. It can be seen that the variance of DSSL is the lowest in both the 60-label and 120-label cases, which indicates that DSSL is more stable than the other baseline methods. In addition, the variance of the model with 120 labels is generally smaller than that of the case with 60 labels, suggesting that the increase in the number of labeled samples improves the stability of the SSL algorithm.

## 5. Discussion

In this study, two encoders are used to learn different features of the sample for predicting different degrees of cognitive impairment: AD, MCI, and NC. With the ADNI neuropsychological dataset and a small number of labels, DSSL achieved an accuracy of 85.47% in the 60-label case and 88.40% in the 120-label case. The comparison results in Table 5 show that our proposed semi-supervised method outperforms the existing semi-supervised methods in terms of accuracy, sensitivity, specificity, recall, and F1-score. The comparison results in Table 6 show that our proposed algorithm is more stable than the existing semi-supervised methods.

Feature selection has an essential role as a precursor to the classification task. PCC is one of the most typical and popular similarity measures. The reason we chose PCC for feature selection is that PCC has the property that shifts in the position and scale of the variable do not cause a change in this coefficient. This property allows the correlation between the neuropsychological test scores after normalization and the diagnosis to be the same as the original values. It helps to improve classification performance while providing physicians with biomarker references for clinical diagnosis. As seen in Table 2, CDR, MMSE, ADAS, and FAQ have strong correlations with the degree of cognitive impairment and their total scores correlate more strongly with the diagnostic outcome compared to the sub-scores.

For computational complexity, Table 5 shows the training time for DSSL and other comparative methods. It can be seen that MixMatch and FixMatch take the shortest time, and our proposed method takes a little longer because it requires updating the parameters of both encoders. All the experiments are performed on a PC with 2.0 GHz, 8-core CPU, and 8 GB RAM on a Windows 10 operating system. Overall, all experiments applying neuropsychological test data for training require less than 3 min, which demonstrates the usability of the proposed method for clinical applications.

The confidence threshold seriously affects the quality of the generated pseudo-labels. Although we find the optimal value of τ in Table 3 through extensive experiments, this is time-consuming, and there is no guarantee that the set threshold will work for each data division. The question to be considered is how to weigh the number of unlabeled samples exceeding the threshold and the consistency rate of pseudo-labels with valid labels. Perhaps automatic learning of this parameter using neural networks would be a better approach. This is also how the model will be improved in the future. DSSL diagnoses AD by using two encoders to learn different features of the sample. To facilitate the visualization of the learned feature representations, we use Shapley values [45] to quantify the importance of features in the algorithm predictions. We sort each feature in the feature representation by its contribution to the model output. Figure 7 shows the top 10 features with the highest contribution in each of the two feature representations, where class 2, 1, and 0 denote AD, MCI, and NC, respectively. As seen in the figure, all features have higher impact scores for AD and NC, while MCI as an intermediate stage is weakly influenced by these features. Moreover, the same features in the two feature representations do not contribute consistently to the algorithm output, which indicates that the two encoders in the proposed method do learn different feature representations. However, there are still limitations in the medical interpretation of these features in correlation with disease pathology. Using expert knowledge to correct the learned feature representation may yield better classification results.

## 6. Conclusions

To accurately determine AD severity with easily available features and a limited number of labels, we propose a novel semi-supervised framework, namely DSSL. We first collect 64 itemized scores from seven neuropsychological tests and use PCC for feature selection. A total of 15 features most relevant to the diagnostic results are selected to serve as input for subsequent semi-supervised experiments. Then, the DSSL model is proposed to better screen for AD and MCI using only neuropsychological tests and a small amount of labeling, without the need for costly PET and MRI, etc. The model uses two encoders and difference regularization to learn two different features from the same sample. Finally, we empirically demonstrate the validity and stability of our method through extensive comparisons with a large number of existing semi-supervised algorithms in terms of accuracy, sensitivity, specificity, recall, F1-score, and variance.

In the future, the proposed algorithm will be applied to other AD biomarkers of multimodal data such as MRI, PET, etc. It would be a promising research direction to use other deep neural network models as encoders to extract potential feature representations of the data and to explore medical interpretations of the relationship between feature representations and disease pathology.

## Figures and Tables

**Figure 1 brainsci-13-00306-f001:**
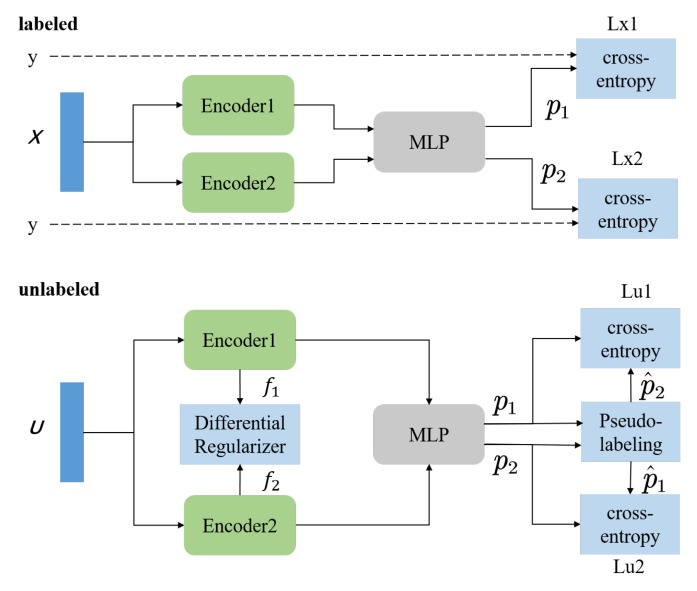
Overview of the proposed model.

**Figure 2 brainsci-13-00306-f002:**
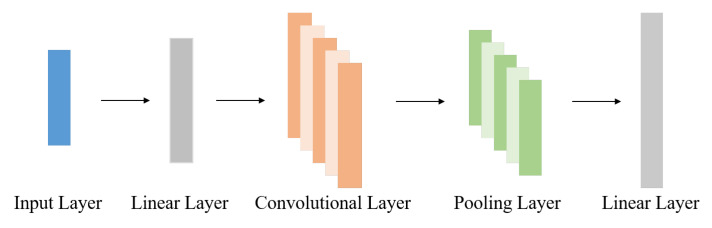
An illustration of the composition of the encoder. The encoder mainly consists of a connection layer, a convolutional layer, a pooling layer, and a connection layer. The main alteration part of different encoders is in the pooling layer.

**Figure 3 brainsci-13-00306-f003:**
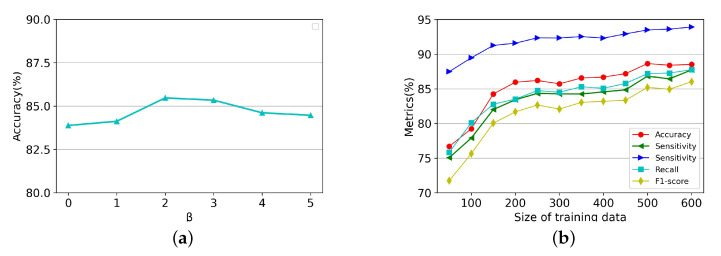
Illustrating (**a**) classification accuracy of the proposed method on different values of β in the 60-label case and (**b**) classification performance of the proposed method on different sizes of training data.

**Figure 4 brainsci-13-00306-f004:**
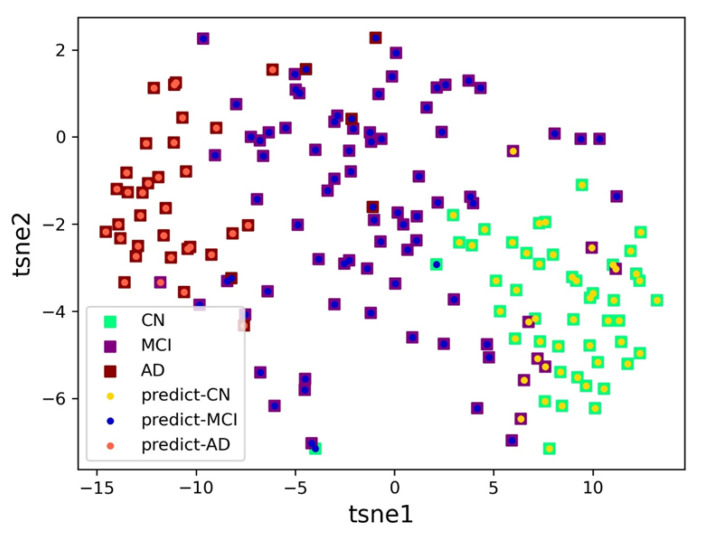
The two-dimensional t-SNE plot of the prediction results of DSSL for the test set samples.

**Figure 5 brainsci-13-00306-f005:**
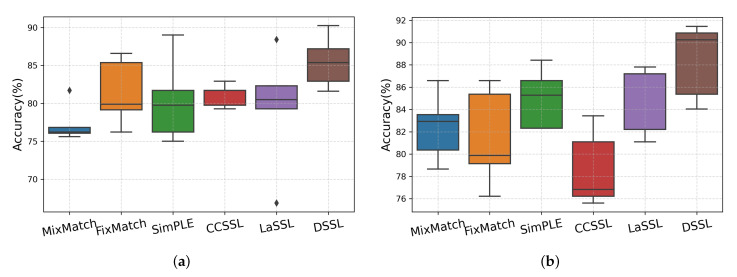
Illustrating (**a**) each model’s accuracy in a 5-fold cross-validation experiment in the 60-label case and (**b**) each model’s accuracy in a 5-fold cross-validation experiment in the 120-label case.

**Figure 6 brainsci-13-00306-f006:**
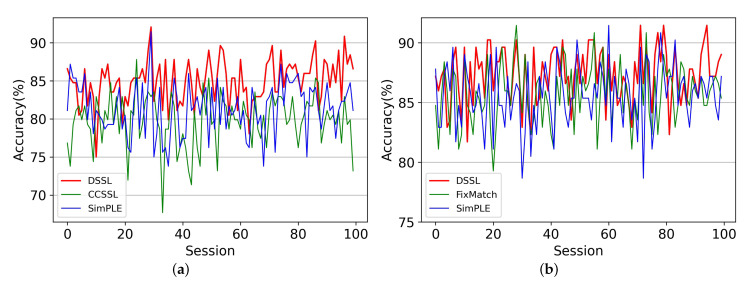
Illustrating (**a**) line graphs of the three methods with a minor variance in the results of 100 experiments in the 60-label case and (**b**) line graphs of the three methods with a minor variance in the results of 100 experiments in the 120-label case.

**Figure 7 brainsci-13-00306-f007:**
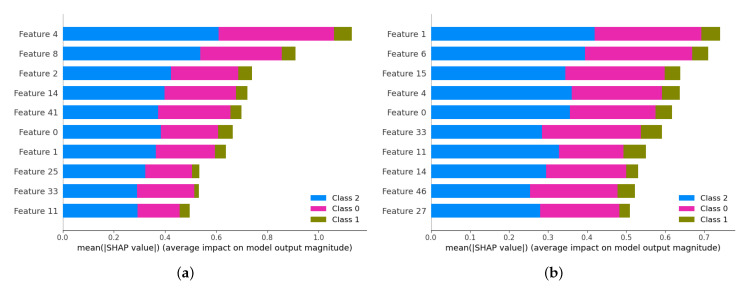
Illustrating (**a**) 10 features with the highest contribution in the feature representation learned by Encoder1 and (**b**) 10 features with the highest contribution in the feature representation learned by Encoder2.

**Table 1 brainsci-13-00306-t001:** Subject characteristics.

Characteristic	AD (*n* = 188)	MCI (*n* = 402)	NC (*n* = 229)	*p*-Value
Gender (M/F)	99/80	257/143	119/110	-
Age	75.3 ± 7.5	74.8 ± 7.4	75.9 ± 5.0	-
MMSE	23.3 ± 2.0	27.0 ± 1.8	29.1 ± 1.0	<0.001
CDR	0.7 ± 0.3	0.5	0	<0.001
FAQ	13.1 ± 6.8	3.9 ± 4.5	0.1 ± 0.6	<0.001
ADAS1	6.1 ± 1.5	4.6 ± 1.4	2.9 ± 1.1	<0.001
RAVLT	23.2 ± 7.7	30.6 ± 9.0	43.3 ± 9.0	<0.001
NPIQ	3.5 ± 3.4	1.9 ± 2.7	0.3 ± 0.9	<0.001
GDS	1.7 ± 1.4	1.6 ± 1.4	0.8 ± 1.1	0.14

Data are expressed as mean ± standard deviation. MMSE = mini-mental state examination, CDR = clinical dementia rating, FAQ = functional activity questionnaire, ADAS1 = word list non-learning (mean) RAVLT = Anterograde episodic memory-verbal, NPIQ = neuropsychiatric inventory Q, GDS = geriatric depression scale. The *p*-values for the differences between AD, MCI and NC are based on two-way *t*-tests with Bonferroni correction.

**Table 2 brainsci-13-00306-t002:** The 15 items with the highest absolute PCC.

Feature Name	Absolute PCC	Feature Name	Absolute PCC
CDR-SB	0.827928	FAQFORM	0.647591
MMSETOTAL	0.766936	RAVLT_immediate	0.629216
ADASMOD	0.743978	FAQFINAN	0.620305
ADAS_Q4	0.721948	FAQTRAVL	0.595336
FAQTOTAL	0.691619	FAQSHOP	0.574347
ADAS11	0.691199	RAVLT_perc_forgetting	0.561252
FAQREM	0.657881	FAQMEAL	0.550885
ADAS_Q1	0.656050		

CDR-SB = CDR sum of boxes, MMSETOTAL = total score of MMSE, FAQTOTAL = total score of FAQ, ADASMOD = total score of ADAS-Cog including Q4 and Q14, ADAS_Q4 = ADAS delayed word recall, FAQTOTAL = total score of FAQ, ADAS11 = total score of ADAS-Cog excluding Q4 and Q14, FAQREM = FAQ remember appointments, ADAS_Q1 = ADAS word recall, FAQFORM = complete forms, RAVLT_immediate = RAVLT immediate recall, FAQFINAN = FAQ manage finance, FAQTRAVL = FAQ travel out of the neighborhood, FAQSHOP = FAQ shop, RAVLT_perc_forgetting = RAVLT Percent Forgetting, FAQMEAL = prepare a balanced meal.

**Table 3 brainsci-13-00306-t003:** Passing rate, impurity rate, and accuracy of test set for DSSL with different thresholds in the 60-label case.

τ	Passing Rate	Impurity Rate	Accuracy
0.25	100	18.55	82.05
0.5	100	16.48	84.12
0.75	99.67	16.45	85.1
0.85	98.95	16.58	84.24
0.9	97.79	15.33	85.47
0.95	95.31	14.32	85.22
0.97	93.36	13.16	85.47
0.99	87.64	10.88	85.22

**Table 4 brainsci-13-00306-t004:** Test set evaluation results for DSSL with different encoders in the 60-label case.

	Pooling Layer	ACC (%)	SEN (%)	SPE (%)	REC (%)	F1 (%)
	Max + Max	84.49	82.42	83.05	91.29	80.46
60-label	Avg + Avg	84.00	82.05	83.00	91.19	80.42
	Max + Avg	85.47	83.77	84.14	91.82	81.92
	Max + Max	88.15	85.89	86.06	93.03	84.60
120-label	Avg + Avg	88.27	86.72	87.37	93.41	85.50
	Max + Avg	88.40	86.99	87.07	93.20	85.53

Max + Max means the pooling layer of Encoder1 is max pooling, and the pooling layer of Encoder2 is also max pooling. The other pooling layers are similar.

**Table 5 brainsci-13-00306-t005:** The semi-supervised evaluation results of each model for the ADNI database data in the 60-label and 120-label cases.

	Method	ACC (%)	SEN (%)	SPE (%)	REC (%)	F1 (%)	Training Time (Minute)
	MixMatch [25] (2019)	77.29	75.40	88.21	76.73	72.40	1.27
	FixMatch [27] (2020)	81.44	79.01	90.27	80.29	76.90	1.15
	SimPLE [34] (2021)	80.34	77.39	89.66	78.80	75.26	2.53
60-label	CCSSL [35] (2022)	81.07	79.74	89.99	80.05	77.11	2.48
	LaSSL [36] (2022)	79.47	76.06	89.15	78.49	74.26	2.82
	DSSL	85.47	83.77	84.14	91.82	81.92	2.42
	MixMatch [25] (2019)	82.42	80.55	91.13	81.57	78.16	1.24
	FixMatch [27] (2020)	84.49	82.10	91.80	83.71	80.46	1.11
	SimPLE [34] (2021)	84.98	82.94	91.92	83.60	85.15	2.51
120-label	CCSSL [35] (2022)	78.64	76.03	88.83	76.92	73.23	2.42
	LaSSL [36] (2022)	85.10	82.63	91.60	83.66	81.07	2.85
	DSSL	88.40	86.99	87.07	93.20	85.53	2.27

**Table 6 brainsci-13-00306-t006:** The variance of the results of each model after 100 experiments in the 60 and 120 label cases.

Method	60 Labels	120 Labels
MixMatch [25] (2019)	3.84	2.87
FixMatch [27] (2020)	3.62	2.48
SimPLE [34] (2021)	3.41	2.62
CCSSL [35] (2022)	3.38	3.38
LaSSL [36] (2022)	4.34	3.07
DSSL	2.91	2.30

## Data Availability

We using open datasets to tested our method. The ADNI dataset can be found in https://adni.loni.usc.edu/ (accessed on 18 December 2021).

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
