# Peer review of "Dual Semi-Supervised Learning for Classification of Alzheimer’s Disease and Mild Cognitive Impairment Based on Neuropsychological Data"

_brainsci, 2023, doi:10.3390/brainsci13020306_

Round 1

Reviewer 1 Report

In this paper, the authors tackled an important problem of automated classification based on deep learning on Alzheimer’s Disease / Mild Cognitive data. The topic is certainly worthy of investigation, and it is widely researched in the literature. The introduction is well written and the results are clearly presented.
However, the manuscript suffers from the following shortcomings in the discussion  which should be, in my opinion, addressed:
-I encourage the authors to rework the discussion part of the manuscript in order to provide more critical review of the current state of the art.
-It would be useful to discuss the advantages and shortcomings of deep learning approaches to clearly present the research gaps which are still to be addressed by the community.
-Also, it would help better present the contribution behind the work reported in this paper.

-Authors should describe the choice of feature selection technique: why Pearson correlation? How many advantages/disadvantages?
-Moreover, please discuss the limitations of the proposed algorithm in detail.

Reviewer 2 Report

This work introduced a semi-supervised method using neuropsychological test scores and scarce labeled data, which introduces difference regularization and consistency regularization with pseudo-labeling. The experimental results show that DSSL achieves the best accuracy and stability in classifying AD, MCI, and NC, in comparison with other approaches.

1. I would suggest providing some examples to visualize the feature representation learned by the classification model. This can help readers better understand how the method works on feature learning. The authors can provide an interpretation of the relationship between the feature representation and the pathology of the disease.

2. It would be interesting to examine the how changing the size of training data would affect the classification performance.

3. How about the computational efficiency of the proposed method? The authors may give an experimental comparison and brief discussion on this point.

4. The authors may briefly discuss the potential limitations of the proposed method and what are the future research directions of this study. How other researchers can work on your study to continue this line of research?

5. Recent studies have suggested the promise of using neuroimaging data for the diagnosis and prognosis of Alzheimer's disease. I would suggest reviewing more of those studies, such as, Modern views of machine learning for precision psychiatry; Sparse interpretation of graph convolutional networks for multi-modal diagnosis of Alzheimer's disease; Interpretable graph convolutional network of multi-modality brain imaging for Alzheimer's disease.

Reviewer 3 Report

I find this a very interesting topic since the proposed system sounds so great. I think the authors should have discussed the various techniques in greater detail in the paper. 

1- Conclusions: The paper's results are not entirely consistent with those in the conclusion.

2- Can you please explain more about this paper's main contribution?  

3- The results and discussions section should be reorganized in a more highlighted, argumentative way. I strongly recommend adding a comparison to some recent studies.

4- How did the authors optimize the neural network's parameters?

5- There should be a comparison between the proposed methods and the most recent articles. It might be better to use at least 4 articles published in the past two years instead of the old articles the authors used. 

Round 2

Reviewer 1 Report

The authors have satisfactorily addressed most of my concerns.

Reviewer 2 Report

The authors have addressed my concerns. The revision is looking good and can be considered for publication.

Reviewer 3 Report

In my opinion, the revised paper has been thoroughly revised in response to the reviewer's comments.